# Bacterial Succession Pattern during the Fermentation Process in Whole-Plant Corn Silage Processed in Different Geographical Areas of Northern China

Chao Wang [1,†], Hongyan Han [2,†], Lin Sun [1], Na Na [1], Haiwen Xu [3], Shujuan Chang [4], Yun Jiang [5] and Yanlin Xue [1,*]

1 Inner Mongolia Engineering Research Center of Development and Utilisation of Microbial Resources in Silage, Inner Mongolia Academy of Agriculture and Animal Husbandry Science, Hohhot 010031, China; wangchao200612@hotmail.co.jp (C.W.); linsun@cau.edu.cn (L.S.); nana13684752695@hotmail.com (N.N.)
2 State Key Laboratory of Reproductive Regulation and Breeding of Grassland Livestock, School of Life Sciences, Inner Mongolia University, Hohhot 010070, China; hanhongyan1018@Outlook.com
3 College of Foreign Languages, Inner Mongolia University of Finance and Economics, Hohhot 010070, China; xhw@imufe.edu.cn
4 Inner Mongolia Key Laboratory of Remote Sensing of Grassland and Emergency Response Technic, Inner Mongolia Forestry and Grassland Monitoring and Planning Institute, Hohhot 010020, China; schang9@emich.edu
5 Department of Animal Sciences, University of Florida, Gainesville, FL 32611, USA; jiangyun0110@ufl.edu
* Correspondence: xueyanlin0925@Outlook.com; Tel.: +86-471-5295628
† These authors have contributed equally to this study and share first authorship.

**Abstract:** Whole-plant corn silage is a predominant forage for livestock that is processed in Heilongjiang province (Daqing city and Longjiang county), Inner Mongolia Autonomous Region (Helin county and Tumet Left Banner) and Shanxi province (Taigu and Shanyin counties) of North China; it was sampled at 0, 5, 14, 45 and 90 days after ensiling. Bacterial community and fermentation quality were analysed. During fermentation, the pH was reduced to below 4.0, lactic acid increased to above 73 g/kg DM ($p < 0.05$) and *Lactobacillus* dominated the bacterial community and had a reducing abundance after 14 days. In the final silages, butyric acid was not detected, and the contents of acetic acid and ammonia nitrogen were below 35 g/kg DM and 100 g/kg total nitrogen, respectively. Compared with silages from Heilongjiang and Inner Mongolia, silages from Shanxi contained less *Lactobacillus* and more *Leuconostoc* ($p < 0.05$), and had a separating bacterial community from 14 to 90 days. *Lactobacillus* was negatively correlated with pH in all the silages ($p < 0.05$), and positively correlated with lactic and acetic acid in silages from Heilongjiang and Inner Mongolia ($p < 0.05$). The results show that the final silages had satisfactory fermentation quality. During the ensilage process, silages from Heilongjiang and Inner Mongolia had similar bacterial-succession patterns; the activity of *Lactobacillus* formed and maintained good fermentation quality in whole-plant corn silage.

**Keywords:** whole-plant corn silage; bacterial community; succession pattern; fermentation quality; fermentation process

## 1. Introduction

Ensiling fresh crops is an advanced technology for storing forage and supplying higher-quality forage to livestock throughout the year [1–4]. Compared with hay, silage exhibits greater dry-matter digestibility, metabolizable energy and crude-protein content, which contribute to improved liveweight gain [5].

In recent decades, whole-plant corn silage has become the predominant forage for the global dairy industry [6]. The characteristics of whole-plant corn silage are good fermentation quality and high energy, along with physically effective fibre, lower harvesting costs, minimised risks of production, elevated yield per area and flexibility to harvest corn

for forage or grain [6–9]. In addition, more than 133 million dairy cattle globally consume about 665 million tons of silage each year, and whole-plant corn silage accounts for more than 40% of forage in dairy cattle farms [10,11].

Previous studies focused on the effects of the growing location and storing temperature on microbial communities and/or succession in whole-plant corn silage [2,12,13]. Gharechahi et al. [12] studied the dynamics of bacterial communities during the ensilage process in whole-plant corn silages obtained from Gorgan (temperate), Isfahan (warm and dry) and Qazvin (cold and dry) in Iran. Guan et al. [2] found different microbial communities in whole-plant corn that originated from five major ecological areas of the provinces of Sichuan, Chongqing and Guizhou in Southwest China. Additionally, high temperature affected the dynamics of the bacterial community in whole-plant corn silage, and resulted in a shift from a homofermentative to a heterofermentative lactic acid bacteria (LAB) population [13]. Moreover, most researchers are interested in the dynamics of microbial communities during the fermentation process in whole-plant corn silage with or without inoculants. Sun et al. [9] reported on bacterial succession in whole-plant corn silage during the initial aerobic, intense fermentation and stable phases. The study also found the LAB fermentation relay during the fermentation process, which was reflected by *Weissella*, *Lactococcus* and *Leuconostoc* in the first 5 h; *Weissella*, *Lactococcus*, *Leuconostoc*, *Lactobacillus* and *Pediococcus* between 5 and 24 h; *Lactobacillus* from 24 h to 60 days [9]. In addition, other studies revealed that adding LAB inoculants could promote bacterial-community succession, improve fermentation quality, and contribute to the accumulation of metabolites in whole-plant corn silage [11,14–17].

Heilongjiang, Inner Mongolia and Shanxi were among the top five provinces regarding whole-plant corn silage production in China in 2016 [18]. In addition, Heilongjiang has a temperate monsoon climate, and Inner Mongolia and Shanxi have a temperate continental climate. We hypothesised that there were differences in bacterial-succession patterns in whole-plant corn silage from different geographical locations. Thus, the objectives of this study were to determine changes in bacterial communities during the fermentation process in whole-plant corn silages processed in Heilongjiang, Inner Mongolia and Shanxi of North China.

## 2. Materials and Methods

### 2.1. Materials and Silage Preparation

The corn (*Zea mays* L.) plants used for ensiling were raised in experimental farms at 6 locations in 3 areas of North China: Heilongjiang province (Daqing city (H_D; 124°43′42.074″ E, 46°18′32.083″ N) and Longjiang county (H_L; 123°7′32.120″ E, 47°21′23.396″ N), cold and wet agricultural area), Inner Mongolia Autonomous Region (Helin county (I_H; 111°36′21.625″ E, 40°28′47.672″ N) and Tumet Left Banner (I_T; 111°9′31.399″ E, 40°41′29.832″ N), temperate and dry pastoral area) and Shanxi province (Taigu (S_T; 112°37′53.792″ E, 37°25′57.230″ N) and Shanyin (S_S; 112°52′06.676″ E, 39°32′54.503″ N) counties, temperate and dry agricultural area) (Figure 1). The variety of corn was 23 Yu for ensiling (no. 2008022, Henan Dajingjiu Seed Industry Co., Ltd., Shangqiu, China). Corn was harvested from 3 fields as replicates in each location; the harvesting stage according to local tradition was 1/3, 1/2 and 1/2 milk-line stage in Heilongjiang, Inner Mongolia and Shanxi, respectively. After harvesting, fresh forage from each field was chopped into 1 to 2 cm pieces using a chaff cutter (Hongguang Industry &Trade Co. Ltd., Ningbo Zhejiang, China), uniformly mixed and randomly divided into 5 batches (500 g per batch) that were ensiled in 5 plastic bags sealed with a vacuum sealer. Silage bags (500 g per bag) were stored at ambient temperature (22 °C to 25 °C) and sampled at 0, 5, 14, 45 and 90 days after ensiling.

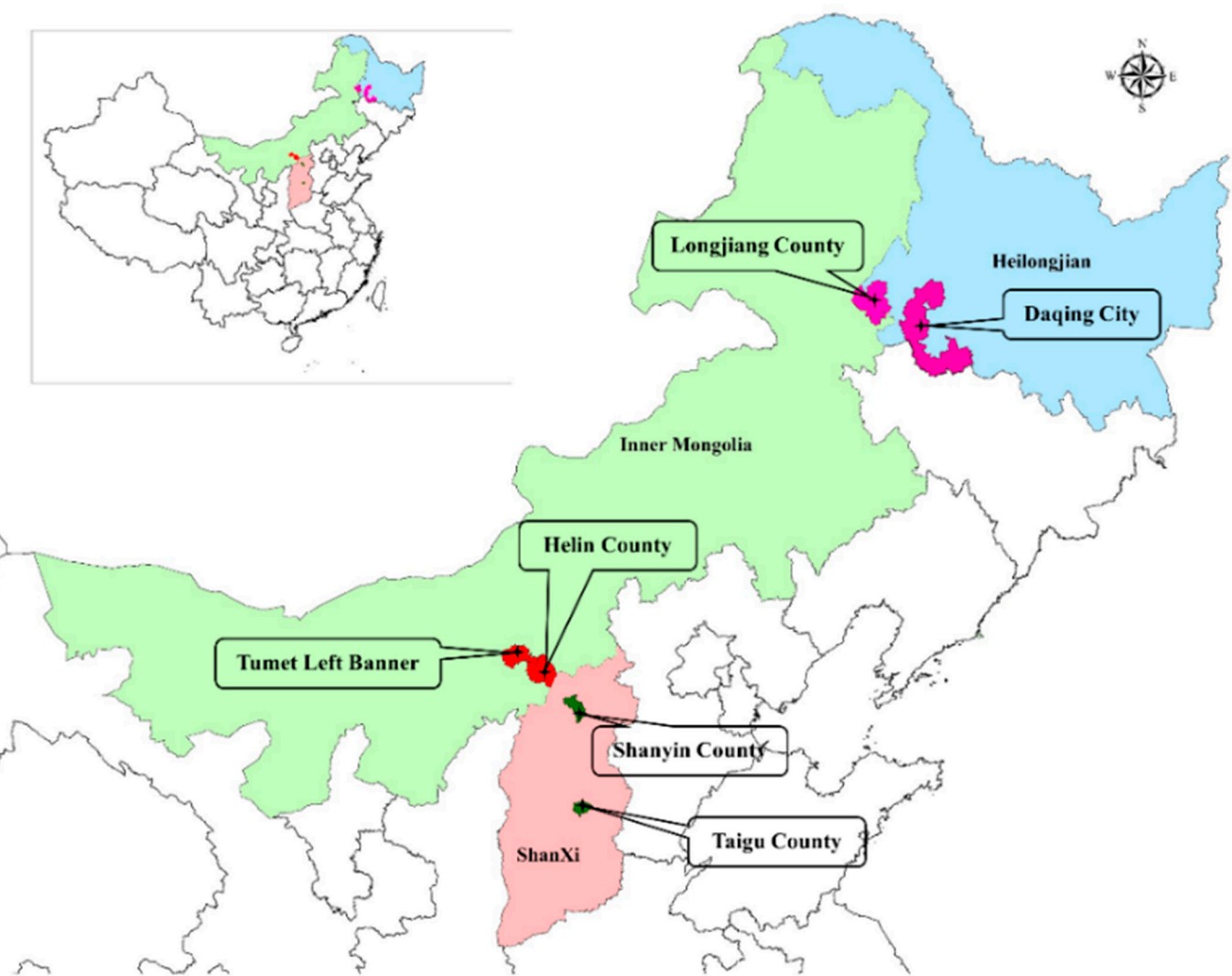

**Figure 1.** Sampling locations of whole-plant corn. Heilongjiang province (Daqing city (124°43′42.074″ E, 46°18′32.083″ N) and Longjiang county (123°7′32.120″ E, 47°21′23.396″ N), cold and wet agricultural area), Inner Mongolia Autonomous Region (Helin county (111°36′21.625″ E, 40°28′47.672″ N) and Tumet Left Banner (111°9′31.399″ E, 40°41′29.832″ N), temperate and dry pastoral area) and Shanxi province (Taigu (112°37′53.792″ E, 37°25′57.230″ N) and Shanyin (112°52′06.676″ E, 39°32′54.503″ N) counties, temperate and dry agricultural area).

### 2.2. Analyses

#### 2.2.1. Physicochemical Analysis

The dry-matter content was measured as follows: drying in a forced-air oven (BPG-9240A, Shanghai Yiheng Scientific Instrument Co., Ltd., Shanghai, China) at 65 °C for 48 h, grinding through a 1 mm screen using a mill (FS-6D; Fichi Machinery Equipment Co., Ltd., Jinan Shandong, China) and drying at 105 °C until a constant mass was reached. The silage extract was prepared as follows: a mixture of 20 g fresh silage and 180 mL sterile water was homogenised for 100 s in a flap-type sterile homogeniser (JX-05, Shanghai Jingxin Industrial Development Co., Ltd., Shanghai, China) and filtered through 4 layers of cheesecloth. The pH of silage was established using a pH meter (PB-10, Sartorius, Gottingen, Germany) to measure the silage extract. Concentrations of lactic acid (LA), acetic acid (AA), propionic acid (PA) and butyric acid in the silages were determined by high-performance liquid chromatography (HPLC; 20A; Shimadzu Co., Ltd., Kyoto, Japan; (detector, SPD-20A diode array detector (210 nm); column, Shodex RS Pak KC-811 (50 °C, Showa Denko K.K., Kawasaki, Japan); mobile phase, 3 mM $HClO_4$ (1.0 mL/min)) [19]. Ammonia nitrogen

(AN) and total nitrogen (TN) contents were measured with a Kjeltec autoanalyser (8400; Foss Co., Ltd., Hillerød, Denmark) according to the Kjeldahl method [20].

### 2.2.2. Microbial Analysis

The counts of LAB, enterobacteria, total aerobic bacteria and yeast in silage were determined by culturing on MRS agar, violet red bile agar, nutrient agar and potato dextrose agar, respectively, at 37 °C for 48 h in an incubator (LRH-70, Shanghai Yiheng Science Instruments Co., Ltd., Shanghai, China) [21].

Then, 20 g of silage from each bag was placed into a self-styled bag at each sampling time and stored at −80 °C to analyse the bacterial communities. Bacterial DNA in silage was extracted by the E.Z.N.A. ®®Stool DNA Kit (D4015, Omega Bio-Tek, Inc., Doraville, GA, USA) according to the manufacturer's instructions. Amplification of the V3–V4 region of the bacterial rRNA gene was operated by polymerase chain reaction (PCR) with primers 341F (5′-CCTACGGGNGGCWGCAG-3′) and 805R (5′-GACTACHVGGGTATCTAATCC-3′) as follows: 98 °C for 30 s, followed by 32 cycles of denaturation at 98 °C for 10 s, annealing at 54 °C for 30 s, an extension at 72 °C for 45 s, and a final extension at 72 °C for 10 min [22]. PCR products were purified by AMPure XT beads (Beckman Coulter Genomics, Danvers, MA, USA), quantified by Qubit (Invitrogen, Carlsbad, CA, USA) and sequenced on an Illumina NovaSeq PE250 platform. High-quality clean tags were obtained from raw reads under specific filtering conditions according to fqtrim (version 0.94). Alpha diversity and beta diversity were calculated by QIIME2, the sequence alignment of species annotation was performed by BLAST, and the alignment databases were SILVA and NT-16S. The stacked bars of bacterial genera were made by Excel (Microsoft 365, Microsoft Corporation, Seattle, DC, USA) according to the relative abundance of the bacterial community. Differences in bacterial communities among sampling areas at each sampling time were analysed using the Mann–Whitney U and Kruskal–Wallis tests by R version 3.6.1. Principal-component analysis (PCA) of the bacterial communities among sampling areas at each sampling time was performed with R version 3.5.0 using OmicStudio tools at https://www.omicstudio.cn/tool (accessed on 10 February 2021).

### 2.3. Statistical Analyses

Data on the dry matter, fermentation quality, microbial counts, bacterial sequencing, and alpha diversity of whole-plant corn silage were analysed as a 3 × 5 factorial design. The model included the effects of the sampling area, sampling time, and their interaction. Differences among sampling locations and sampling times were analysed by one-factor analysis of variance using the general linear model (GLM) procedure of SAS (version 9.1.3; SAS Inst. Inc., Cary, NC, USA). The interaction of sampling area and sampling time was analysed using the PDIFF procedure of SAS. Differences were compared with the least significant difference, and significance was declared at $p \leq 0.05$. Correlation between bacterial community (top 10 genera) and fermentation quality (pH, LA, AA and PA) was established with R 3.6.1.

## 3. Results

### 3.1. Fermentation Quality and Microbial Counts

Sampling area influenced pH and contents of AA, PA and AN ($p < 0.05$); sampling time affected pH, LA, AA, PA and AN ($p < 0.05$). Moreover, pH, AA and AN were interactively influenced by sampling area and sampling time ($p < 0.05$; Table 1). The counts of LAB, enterobacteria and total aerobic bacteria were affected by the sampling area ($p < 0.05$), and the sampling time influenced LAB, enterobacteria, total aerobic bacteria and yeast ($p < 0.05$), which were also interactively impacted by the sampling area and sampling time ($p < 0.05$; Table 2). No butyric acid or mould were detected in the silages.

**Table 1.** Dry matter (g/kg), pH, organic acid contents (g/kg dry matter) and ammonia nitrogen (g/kg total nitrogen) of whole-plant corn silages during fermentation ($n = 3$).

| Items | Day | H_D | H_L | I_H | I_T | S_T | S_S | SEM | p-Value | A | T | A × T |
|---|---|---|---|---|---|---|---|---|---|---|---|---|
| Dry matter | 0 | 254 [c] | 245 [c] | 285 [b] | 288 [b] | 303 [a] | 302 [ABa] | 4.14 | <0.001 | <0.001 | 0.007 | 0.0694 |
| | 5 | 256 [b] | 254 [b] | 312 [a] | 301 [a] | 307 [a] | 303 [ABa] | 8.53 | <0.001 | | | |
| | 14 | 248 [c] | 253 [c] | 289 [b] | 298 [ab] | 304 [ab] | 317 [ABa] | 6.58 | <0.001 | | | |
| | 45 | 256 [b] | 240 [b] | 280 [a] | 302 [a] | 302 [a] | 287 [Ba] | 5.72 | <0.001 | | | |
| | 90 | 257 [d] | 250 [d] | 291 [c] | 294 [c] | 309 [b] | 331 [Aa] | 4.02 | <0.001 | | | |
| | SEM | 4.56 | 5.67 | 6.93 | 5.62 | 6.04 | 7.02 | | | | | |
| | p-value | 0.645 | 0.407 | 0.068 | 0.385 | 0.893 | 0.012 | | | | | |
| pH | 0 | 6.03 [A] | 5.96 [A] | 6.06 [A] | 6.07 [A] | 5.98 [A] | 5.99 [A] | 0.029 | 0.106 | <0.001 | <0.001 | <0.001 |
| | 5 | 3.91 [Ba] | 3.87 [Ba] | 3.74 [Bb] | 3.72 [Bb] | 3.75 [Bb] | 3.85 [Ba] | 0.020 | <0.001 | | | |
| | 14 | 3.87 [BCa] | 3.86 [Ba] | 3.58 [Cc] | 3.62 [Cc] | 3.75 [Bb] | 3.83 [Ba] | 0.018 | <0.001 | | | |
| | 45 | 3.81 [Ca] | 3.78 [Cab] | 3.74 [Bbc] | 3.67 [BCd] | 3.69 [Ccd] | 3.76 [Cab] | 0.016 | <0.001 | | | |
| | 90 | 3.80 [Ca] | 3.62 [Db] | 3.52 [Cc] | 3.53 [Dc] | 3.58 [Db] | 3.60 [Db] | 0.012 | <0.001 | | | |
| | SEM | 0.019 | 0.018 | 0.023 | 0.025 | 0.013 | 0.019 | | | | | |
| | p-value | <0.001 | <0.001 | <0.001 | <0.001 | <0.001 | <0.001 | | | | | |
| Lactic acid | 0 | ND | ND | ND | ND | ND | ND | - | - | 0.148 | <0.001 | 0.092 |
| | 5 | 111 [BCa] | 72.6 [Bb] | 72.9 [Cb] | 93.6 [Bb] | 76.5 [Cb] | 80.8 [Cb] | 4.81 | <0.001 | | | |
| | 14 | 99.6 [Cbc] | 72.8 [Bc] | 92.5 [Bbc] | 174 [Aa] | 100 [Bbc] | 117 [Ab] | 6.48 | <0.001 | | | |
| | 45 | 123 [ABb] | 128 [Ab] | 152 [Aa] | 94.3 [Bc] | 119 [Ab] | 100 [Bc] | 4.67 | <0.001 | | | |
| | 90 | 139 [Aa] | 73.0 [Bc] | 138 [Aa] | 96.0 [Bb] | 122 [Aa] | 100 [Bb] | 4.87 | <0.001 | | | |
| | SEM | 5.35 | 4.46 | 5.07 | 5.06 | 4.03 | 4.13 | | | | | |
| | p-value | <0.001 | <0.001 | <0.001 | <0.001 | <0.001 | <0.001 | | | | | |
| Acetic acid | 0 | ND | ND | ND | ND | ND | ND | - | - | <0.001 | <0.001 | 0.038 |
| | 5 | 23.9 [BCa] | 14.2 [Bb] | 12.0 [Bb] | 6.11 [Ac] | 13.4 [Bb] | 12.9 [Bb] | 1.09 | <0.001 | | | |
| | 14 | 19.8 [Ca] | 10.8 [Bbc] | 10.5 [Bbc] | 8.70 [Ac] | 14.7 [Babc] | 16.8 [ABab] | 1.71 | 0.006 | | | |
| | 45 | 27.2 [Ba] | 32.3 [Aa] | 20.0 [Aab] | 7.54 [Ac] | 23.4 [Aab] | 20.4 [Aab] | 3.80 | 0.012 | | | |
| | 90 | 33.5 [Aa] | 17.5 [Bc] | 21.2 [Abc] | 9.57 [Ad] | 24.0 [Ab] | 18.6 [ABc] | 1.41 | <0.001 | | | |
| | SEM | 1.57 | 4.12 | 0.934 | 1.30 | 0.630 | 1.52 | | | | | |
| | p-value | <0.001 | 0.004 | <0.001 | 0.003 | <0.001 | <0.001 | | | | | |

**Table 1.** *Cont.*

| Items | Day | H_D | H_L | I_H | I_T | S_T | S_S | SEM | *p*-Value | *p*-Value A | *p*-Value T | *p*-Value A × T |
|---|---|---|---|---|---|---|---|---|---|---|---|---|
| Propionic acid | 0 | ND | ND | ND | ND | ND | ND | - | - | <0.001 | <0.001 | 0.076 |
| | 5 | N$^D$ | 3.39 $^{Ba}$ | 2.93 $^{Da}$ | 2.24 $^{Ca}$ | 3.08 $^{Ca}$ | N$^D$ | 0.2880 | <0.001 | | | |
| | 14 | 6.39 $^{Cb}$ | 3.99 $^{Bbc}$ | N$^D$ | 1.22 $^{Cc}$ | 14.6 $^{Ba}$ | N$^D$ | 1.1440 | <0.001 | | | |
| | 45 | 10.9 $^{Bb}$ | 13.9 $^{Ab}$ | 6.17 $^{Bc}$ | 6.86 $^{Bc}$ | 17.3 $^{Ba}$ | 6.41 $^{Bc}$ | 0.9800 | <0.001 | | | |
| | 90 | 16.3 $^{Ab}$ | 11.2 $^{Ac}$ | 9.61 $^{Ac}$ | 10.6 $^{Ac}$ | 20.5 $^{Aa}$ | 10.3 $^{Ac}$ | 0.9965 | <0.001 | | | |
| | SEM | 0.947 | 1.10 | 0.565 | 0.717 | 0.891 | 0.521 | | | | | |
| | *p*-value | <0.001 | <0.001 | <0.001 | <0.001 | <0.001 | <0.001 | | | | | |
| Ammonia nitrogen | 0 | 8.47 $^{Da}$ | 9.70 $^{Ca}$ | 7.67 $^{Ca}$ | 2.59 $^{Db}$ | 5.98 $^{Dab}$ | 10.1 $^{Ca}$ | 1.2479 | 0.010 | <0.001 | <0.001 | <0.001 |
| | 5 | 39.4 $^{Ca}$ | 33.8 $^{Bb}$ | 33.1 $^{Bb}$ | 23.1 $^{Cc}$ | 33.9 $^{Cb}$ | 25.8 $^{Bc}$ | 1.2595 | <0.001 | | | |
| | 14 | 61.4 $^{Ba}$ | 35.2 $^{Bc}$ | 51.0 $^{Ab}$ | 26.4 $^{Bc}$ | 29.0 $^{Bc}$ | 31.5 $^{Bc}$ | 2.5968 | <0.001 | | | |
| | 45 | 99.7 $^{Aa}$ | 62.8 $^{Ab}$ | 50.3 $^{Ac}$ | 36.1 $^{Ad}$ | 59.3 $^{Ab}$ | 46.9 $^{Ac}$ | 1.9001 | <0.001 | | | |
| | 90 | 91.4 $^{Aa}$ | 66.2 $^{Ab}$ | 50.8 $^{Acd}$ | 36.3 $^{Ae}$ | 61.0 $^{Abc}$ | 44.9 $^{Ade}$ | 3.4829 | <0.001 | | | |
| | SEM | 2.65 | 1.61 | 2.02 | 0.995 | 1.45 | 3.72 | | | | | |
| | *p*-value | <0.001 | <0.001 | <0.001 | <0.001 | <0.001 | <0.001 | | | | | |

Whole-plant corn silages were processed at 6 locations in 3 areas of North China: Heilongjiang province (Daqing city (H_D; 124°43′42.074″ E, 46°18′32.083″ N) and Longjiang county (H_L; 123°7′32.120″ E, 47°21′23.396″ N), cold and wet agricultural area), Inner Mongolia Autonomous Region (Helin county (I_H; 111°36′21.625″ E, 40°28′47.672″ N) and Tumet Left Banner (I_T; 111°9′31.399″ E, 40°41′29.832″ N), temperate and dry pastoral area) and Shanxi province (Taigu (S_T; 112°37′53.792″ E, 37°25′57.230″ N) and Shanyin (S_S; 112°52′06.676″ E, 39°32′54.503″ N) counties, temperate and dry agricultural area). Sampling times were at 0, 5, 14, 45 and 90 days after ensiling. Note: SEM, standard error of the mean; A, sampling area; T, sampling time. Values with different uppercase ($^{A-D}$) and lowercase ($^{a-e}$) letters indicate significant differences among sampling times of each lactation and sampling locations at the same time, respectively. ND, not detected.

**Table 2.** Microbial counts (log colony-forming units/g fresh weight) of whole-plant corn silages during fermentation (*n* = 3).

| Items | Day | H_D | H_L | I_H | I_T | S_T | S_S | SEM | *p*-Value | *p*-Value A | *p*-Value T | *p*-Value A × T |
|---|---|---|---|---|---|---|---|---|---|---|---|---|
| Lactic acid bacteria | 0 | 8.36 $^{Bb}$ | 8.50 $^{Bb}$ | 6.80 $^{Cd}$ | 9.21 $^{Aa}$ | 7.41 $^{Cc}$ | 6.33 $^{Be}$ | 0.076 | <0.001 | 0.047 | <0.001 | 0.012 |
| | 5 | 8.97 $^{Ab}$ | 8.88 $^{Ab}$ | 8.31 $^{Bc}$ | 8.95 $^{Bb}$ | 9.35 $^{Aa}$ | 9.43 $^{Aa}$ | 0.085 | <0.001 | | | |
| | 14 | 8.77 $^{Ab}$ | 8.71 $^{Ab}$ | 8.65 $^{Ab}$ | 7.86 $^{Cc}$ | 8.55 $^{Bb}$ | 9.31 $^{Aa}$ | 0.075 | <0.001 | | | |
| | 45 | 6.49 $^{Cb}$ | 7.45 $^{Ca}$ | 6.65 $^{Cb}$ | 6.59 $^{Db}$ | 6.10 $^{Db}$ | 6.61 $^{Bb}$ | 0.146 | <0.001 | | | |
| | 90 | 5.35 $^{Db}$ | 6.49 $^{Da}$ | 4.75 $^{Dc}$ | 5.16 $^{Eb}$ | 4.05 $^{Ed}$ | 4.13 $^{Cd}$ | 0.071 | <0.001 | | | |
| | SEM | 0.087 | 0.061 | 0.072 | 0.063 | 0.125 | 0.133 | | | | | |
| | *p*-value | <0.001 | <0.001 | <0.001 | <0.001 | <0.001 | <0.001 | | | | | |

**Table 2.** *Cont.*

| Items | Day | H_D | H_L | I_H | I_T | S_T | S_S | SEM | p-Value | A | T | A × T |
|---|---|---|---|---|---|---|---|---|---|---|---|---|
| Enterobacteria | 0 | 6.53 Ad | 8.44 Aa | 7.48 Ac | 7.99 Ab | 7.30 Ac | 6.06 Ae | 0.096 | <0.001 | 0.048 | <0.001 | 0.004 |
| | 5 | ND | ND | ND | ND | ND | ND | - | - | | | |
| | 14 | ND | ND | ND | ND | ND | ND | - | - | | | |
| | 45 | ND | ND | ND | ND | ND | ND | - | - | | | |
| | 90 | ND | ND | ND | ND | ND | ND | - | - | | | |
| | SEM | 0.035 | 0.036 | 0.063 | 0.026 | 0.004 | 0.062 | | | | | |
| | p-value | <0.001 | <0.001 | <0.001 | <0.001 | <0.001 | <0.001 | | | | | |
| Total aerobic bacteria | 0 | 7.81 Bc | 9.12 Ab | 9.13 Ab | 9.50 Aa | 7.64 Cc | 6.91 Cd | 0.091 | <0.001 | 0.009 | <0.001 | <0.001 |
| | 5 | 8.73 Aa | 8.79 Ba | 8.29 Bb | 8.99 Ba | 9.08 Aa | 9.12 Ba | 0.120 | 0.004 | | | |
| | 14 | 8.75 Ab | 8.54 Cc | 8.50 Bc | 7.83 Cd | 8.49 Bc | 9.44 Aa | 0.060 | <0.001 | | | |
| | 45 | 6.19 Cc | 8.53 Ca | 4.68 Ce | 6.67 Db | 6.20 Dc | 5.65 Ed | 0.043 | <0.001 | | | |
| | 90 | 6.22 Cb | 5.61 Dc | 4.78 Cd | 5.42 Ec | 4.39 Ee | 6.69 Da | 0.070 | <0.001 | | | |
| | SEM | 0.054 | 0.655 | 0.081 | 0.092 | 0.115 | 0.066 | | | | | |
| | p-value | <0.001 | <0.001 | <0.001 | <0.001 | <0.001 | <0.001 | | | | | |
| Yeasts | 0 | 5.59 Cd | 5.26 Dd | 7.72 Bb | 8.19 Aa | 6.19 Cc | 6.60 Cc | 0.134 | <0.001 | 0.648 | <0.001 | <0.001 |
| | 5 | 8.09 Ac | 7.25 Bd | 9.22 Aa | 8.19 Ac | 8.73 Ab | 8.46 Bbc | 0.132 | <0.001 | | | |
| | 14 | 7.74 Aa | 7.54 Ab | 7.72 Ba | 7.59 Bab | 7.50 Bb | 7.43 Ab | 0.044 | 0.002 | | | |
| | 45 | 6.15 Bab | 6.55 Ca | 3.77 Ce | 5.46 Cc | 5.97 Cb | 4.77 Dd | 0.133 | <0.001 | | | |
| | 90 | 6.65 Ba | 4.83 Eb | 3.64 Cc | 5.23 Db | 4.73 Db | 5.11 Db | 0.203 | <0.001 | | | |
| | SEM | 0.170 | 0.071 | 0.083 | 0.070 | 0.113 | 0.239 | | | | | |
| | p-value | <0.001 | <0.001 | <0.001 | <0.001 | <0.001 | <0.001 | | | | | |

Whole-plant corn silages were processed at 6 locations in 3 areas of North China: Heilongjiang province (Daqing city (H_D; 124°43′42.074″ E, 46°18′32.083″ N) and Longjiang county (H_L; 123°7′32.120″ E, 47°21′23.396″ N), cold and wet agricultural area), Inner Mongolia Autonomous Region (Helin county (I_H; 111°36′21.625″ E, 40°28′47.672″ N) and Tumet Left Banner (I_T; 111°9′31.399″ E, 40°41′29.832″ N), temperate and dry pastoral area) and Shanxi province (Taigu (S_T; 112°37′53.792″ E, 37°25′57.230″ N) and Shanyin (S_S; 112°52′06.676″ E, 39°32′54.503″ N) counties, temperate and dry agricultural area). Sampling times were at 0, 5, 14, 45 and 90 days after ensiling. Note: SEM, standard error of the mean; A, sampling area; T, sampling time. Values with different uppercase ([A–E]) and lowercase ([a–e]) letters indicate significant differences among sampling times of each lactation and sampling locations at the same time, respectively. ND, not detected.

## 3.2. Bacterial Communities

A total of 5,669,279 clean reads were obtained from 90 samples of whole-plant corn silage according to 16S rRNA. Sampling area and sampling time affected the observed species diversity, Shannon, Simpson and CHAO1 indices ($p < 0.05$), and had an interactive effect on observed species and CHAO1 ($p < 0.05$) (Table 3). According to PCA, bacterial communities in I5_H_1, I5_H_2, I5_H_3 and S5_S_1 were separated from other silages at 5 days after ensiling (Figure 2A). Silages from Shanxi (S14_S and S14_T) and I14_H had separation from other silages at 14 days (Figure 2B); additionally, silages from Shanxi (S_S and S_T) were clearly separated from other silages at 45 and 90 days (Figure 2C,D).

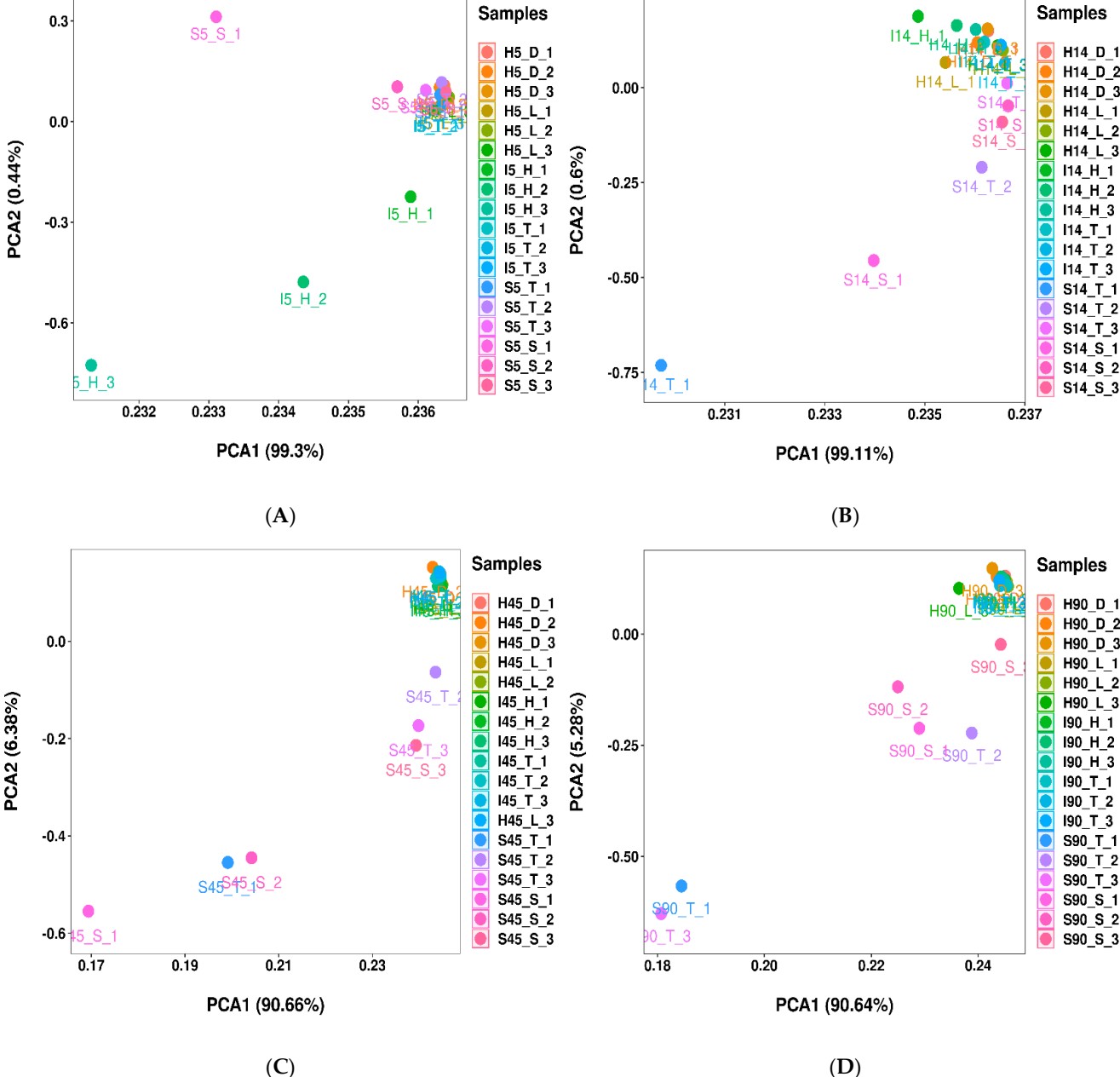

**Figure 2.** Principal-component analysis of bacterial communities at (**A**) 5, (**B**) 14, (**C**) 45 and (**D**) 90 days after ensiling (*n* = 3). Whole-plant corn silages were processed at 6 locations in 3 areas of North China: Heilongjiang province (Daqing city (H_D; 124°43′42.074″ E, 46°18′32.083″ N) and Longjiang county (H_L; 123°7′32.120″ E, 47°21′23.396″ N), cold and wet agricultural area), Inner Mongolia Autonomous Region (Helin county (I_H; 111°36′21.625″ E, 40°28′47.672″ N) and Tumet Left Banner (I_T; 111°9′31.399″ E, 40°41′29.832″ N), temperate and dry pastoral area) and Shanxi province (Taigu (S_T; 112°37′53.792″ E, 37°25′57.230″ N) and Shanyin (S_S; 112°52′06.676″ E, 39°32′54.503″ N) counties, temperate and dry agricultural area).

In silages from Heilongjiang, *Sphingobacterium*, *Stenotrophomonas*, *Sphingomonas* and *Allorhizobium–Neorhizobium–Pararhizobium–Rhizobium* in H_D were the dominant bacterial genera at 0 days (17.4%, 14.8% and 7.61%, respectively), decreased rapidly in the first 5 days, and then stayed at a low level. *Leuconostoc*, *Weissella* and *Sphingobacterium* in H0_L dominated the bacterial community. *Leuconostoc* and *Weissella* in H_L quickly reduced in the first 5 days and then stayed at a low level. The abundance of *Sphingobacterium* in H_L was reduced to 0.319% at 5 days, and then increased to 7.98% at 90 days. Moreover, the abundance of *Lactobacillus* in H_D and H_L was 1.72% and 0.32% at 0 days, increased rapidly to 84.1% and 92.3% at 5 days and was then reduced to 71.23% and 70.4% at 90 days, respectively (Figure 3A).

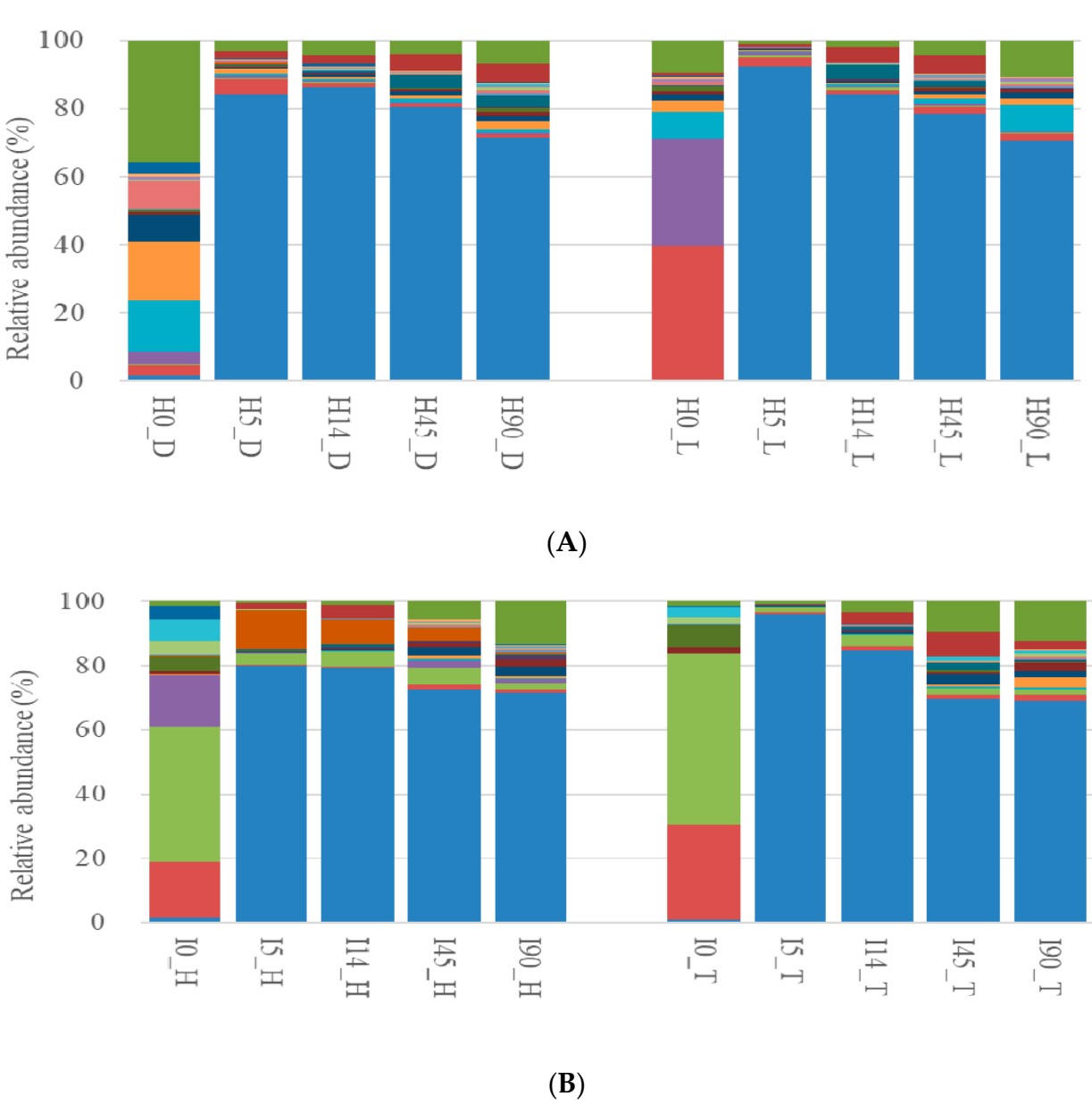

**Figure 3.** *Cont.*

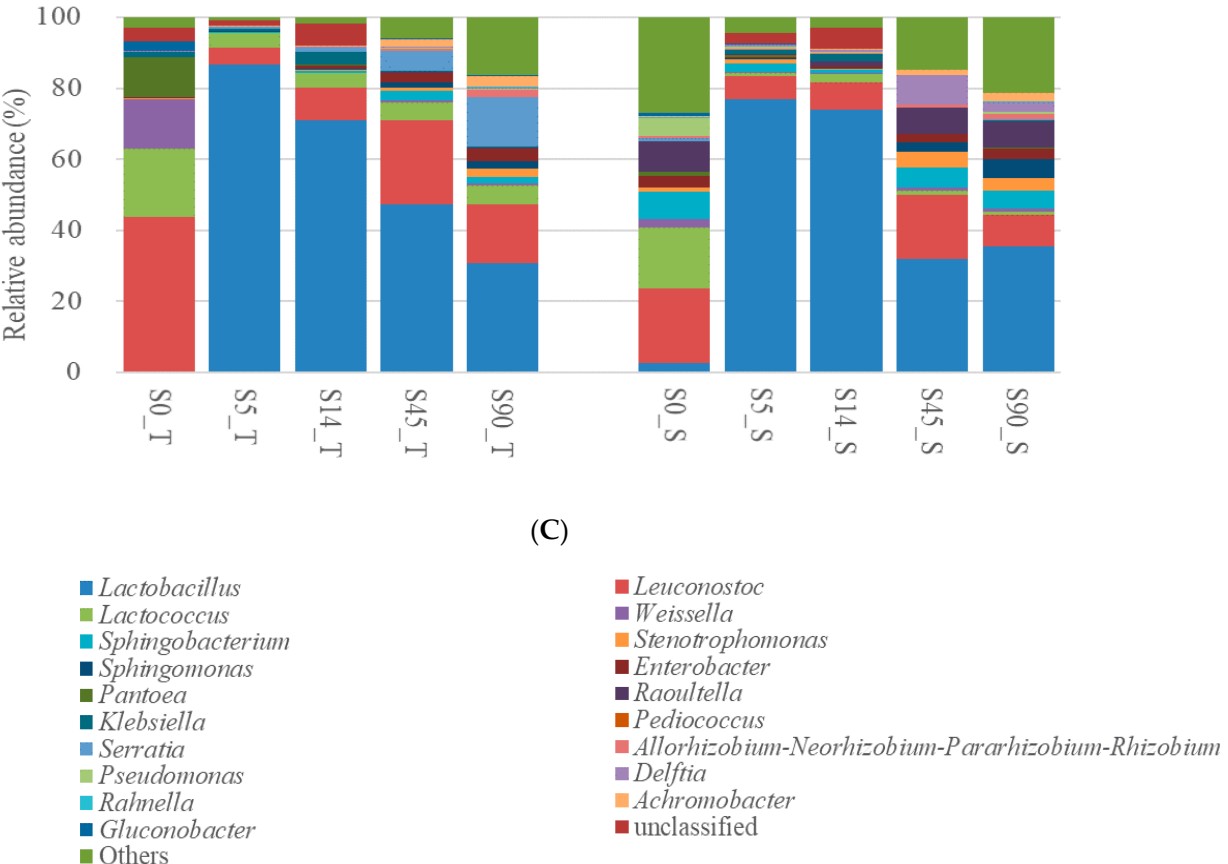

(**C**)

- Lactobacillus
- Lactococcus
- Sphingobacterium
- Sphingomonas
- Pantoea
- Klebsiella
- Serratia
- Pseudomonas
- Rahnella
- Gluconobacter
- Others
- Leuconostoc
- Weissella
- Stenotrophomonas
- Enterobacter
- Raoultella
- Pediococcus
- Allorhizobium-Neorhizobium-Pararhizobium-Rhizobium
- Delftia
- Achromobacter
- unclassified

**Figure 3.** Relative abundance of bacterial communities (genus level) in whole-plant corn silage from (**A**) Heilongjiang, (**B**) Inner Mongolia and (**C**) Shanxi at 0, 5, 14, 45 and 90 days after ensiling (*n* = 3). Whole-plant corn silages were processed at 6 locations in 3 areas of North China: Heilongjiang province (Daqing city (H_D; 124°43′42.074″ E, 46°18′32.083″ N) and Longjiang county (H_L; 123°7′32.120″ E, 47°21′23.396″ N), cold and wet agricultural area), Inner Mongolia Autonomous Region (Helin county (I_H; 111°36′21.625″ E, 40°28′47.672″ N) and Tumet Left Banner (I_T; 111°9′31.399″ E, 40°41′29.832″ N), temperate and dry pastoral area) and Shanxi province (Taigu (S_T; 112°37′53.792″ E, 37°25′57.230″ N) and Shanyin (S_S; 112°52′06.676″ E, 39°32′54.503″ N) counties, temperate and dry agricultural area).

In silages from Inner Mongolia, *Lactobacillus* in I_H and I_T had a rapidly rising abundance in the first 5 days (80.2% and 95.8%, respectively), and it decreased to 71.6% and 69.0% at 90 days, respectively. Abundances of *Leuconostoc* and *Lactococcus* rapidly decreased in the first 5 days, and then maintained low levels in silages. *Weissella* in I_H was quickly reduced from 15.9% at 0 days to 0.20% at 5 days, and then stayed at a low level. *Pediococcus* in I_H increased from 0.32% at 0 days to 12.5% at 5 days, and then reduced to 0.76% at 90 days. No *Weissella* and *Pediococcus* were detected in I_T (Figure 3B).

In silages from Shanxi, the main bacterial genera at 0 days were *Leuconostoc*, *Lactococcus*, *Weissella* and *Pantoea* in S_T, and *Leuconostoc*, *Lactococcus* and *Sphingobacterium* in S_S. The abundance of *Lactobacillus* in S_T and S_S rapidly increased in the first 5 days (86.6% and 77.0%, respectively), and then reduced to 30.8% and 35.6% at 90 days, respectively. *Leuconostoc* in S_T and S_S reduced to 4.90% and 6.32 % at 5 days, increased to 23.68% and 17.85% at 45 days and then decreased to 16.53% and 8.73% at 90 days, respectively. *Lactococcus* in S_T was reduced to 3.95% at 5 days, and then increased to 5.37% at 90 days; in S_S, it was decreased in the first 5 days and then stayed at a low level. *Weissella* was a minor taxon after 5 days in the silages (Figure 3C). H5 and S5 contained more *Leuconostoc* and less *Pediococcus* than those in I5 (*p* < 0.05); additionally, I5 and S5 had higher levels of *Lactococcus* than those in H5 (*p* < 0.05) (Figure 4A). At 14, 45 and 90 days, silages from Heilongjiang and Inner Mongolia contained greater amounts of *Lactobacillus* and less *Leuconostoc* than

those from Shanxi (*p* < 0.05). Moreover, silages from Heilongjiang had lower *Lactococcus* than that of other silages from 14 to 90 days (*p* < 0.05) (Figure 4B–D).

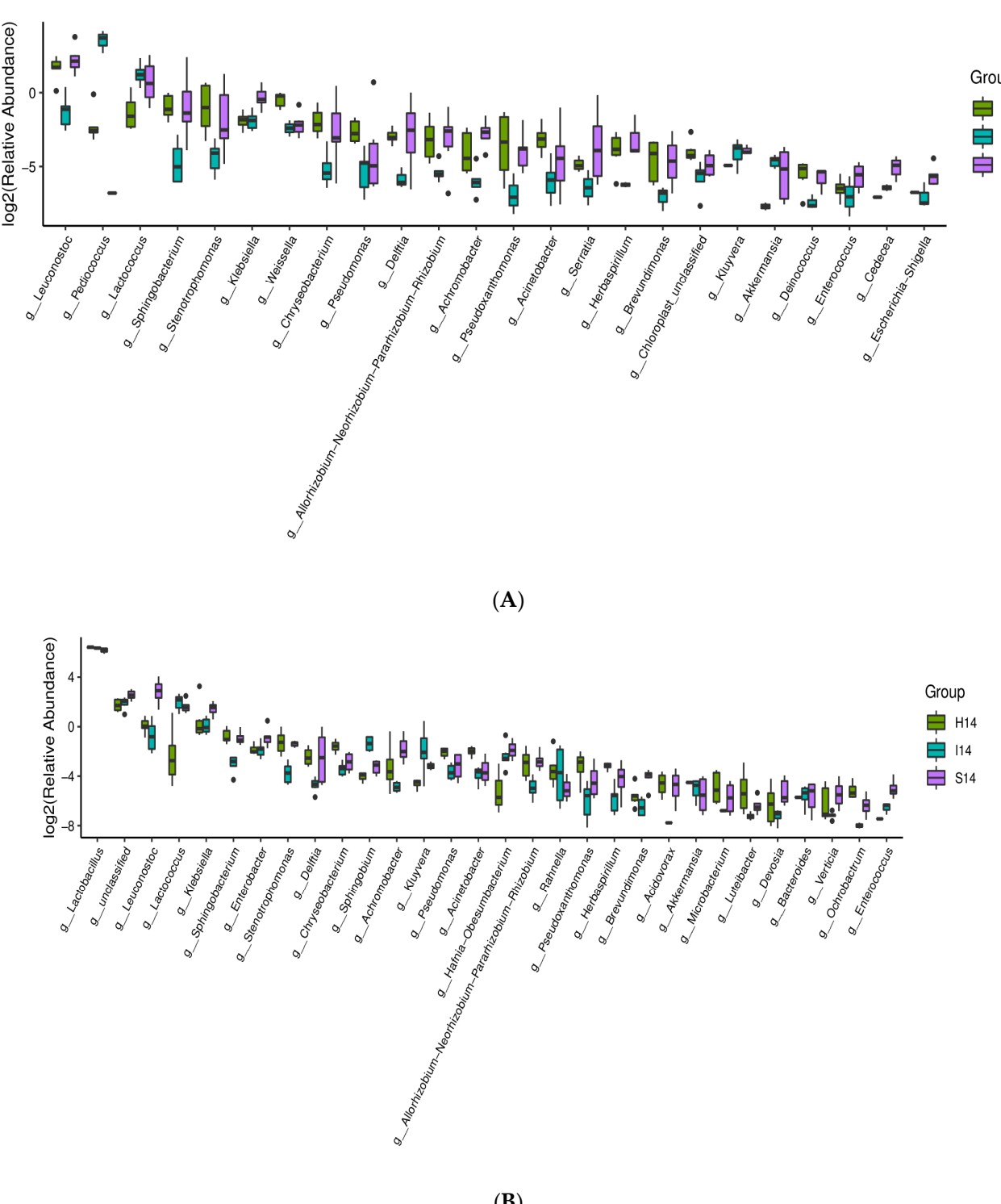

(**A**)

(**B**)

**Figure 4.** *Cont.*

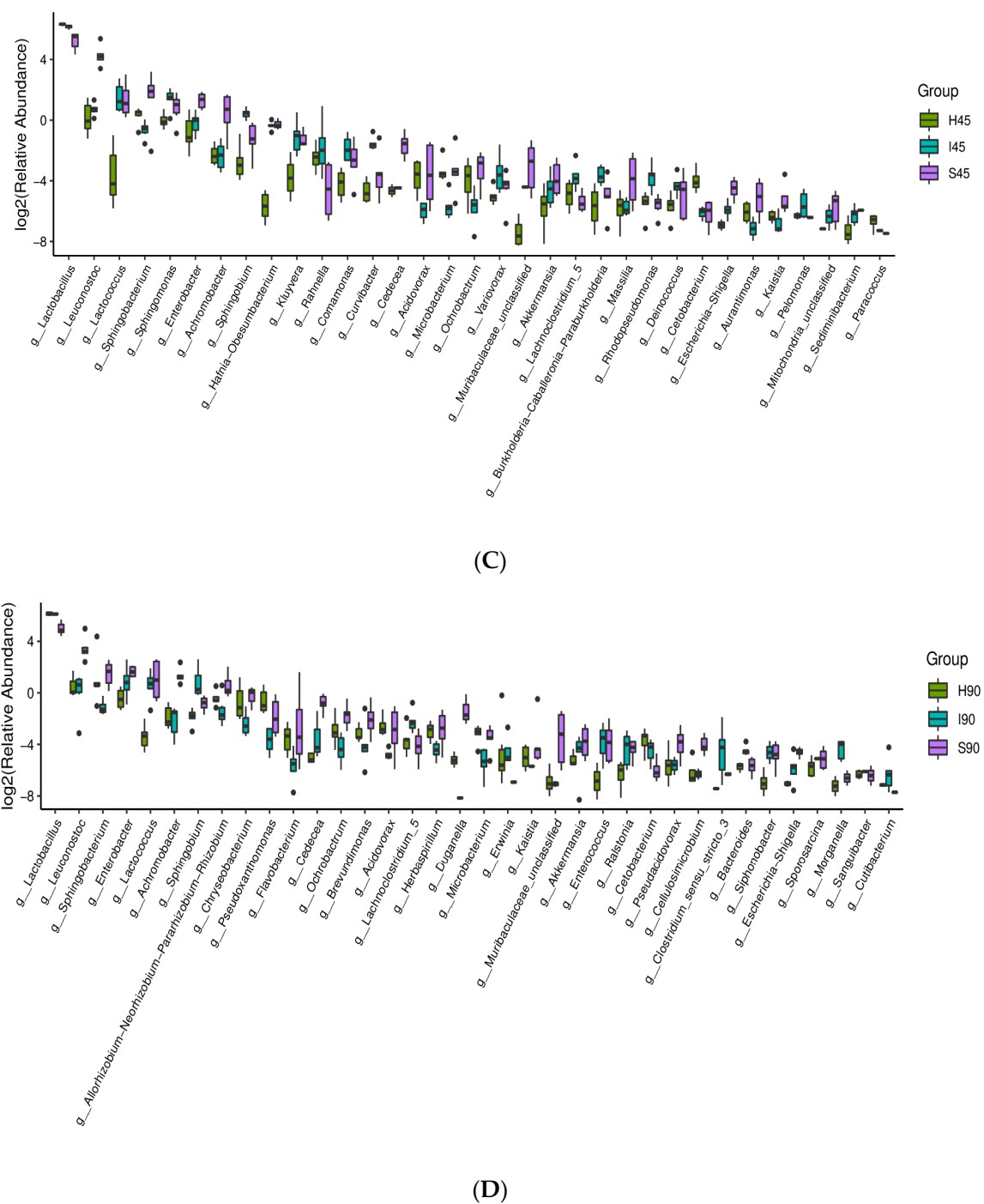

(**C**)

(**D**)

**Figure 4.** Difference in bacterial communities (genus level) among silages from Heilongjiang (H; cold and wet agricultural area), Inner Mongolia (I; temperate and dry pastoral area), and Shanxi (S; temperate and dry agricultural area) at (**A**) 5 and (**B**) 14 days after ensiling (*n* = 6), and (**C**) 45 and (**D**) 90 days after ensiling (*n* = 6).

**Table 3.** Sequencing data and alpha diversity of bacteria and fungi in whole-plant corn silage during fermentation (*n* = 3).

| Items | Day | H_D | H_L | I_H | I_T | S_T | S_S | SEM | *p*-Value | *p*-Value A | T | A × T |
|---|---|---|---|---|---|---|---|---|---|---|---|---|
| Raw reads | 0 | 72672B5 | 77394 | 57874 C | 61165 | 76717 A | 46439 C | 6798 | 0.048 | <0.001 | <0.001 | 0.002 |
| | 5 | 82678 A | 84881 | 82158 AB | 82724 | 82624 A | 84795 A | 1217 | 0.420 | | | |
| | 14 | 83060 A | 85767 | 86449 A | 85417 | 84053 A | 83291 A | 1454 | 0.656 | | | |
| | 45 | 83672 Aa6 | 80723 a | 68678 BCb | 82555 a | 64388 Bbc | 59492 Bc | 1995 | <0.001 | | | |
| | 90 | 80527 Aa | 69953 ab | 63394 Cbc | 74775 ab | 56173 Bc | 53520 BCc | 3522 | 0.001 | | | |
| | SEM | 1768 | 3461 | 4412 | 6015 | 3256 | 2531 | | | | | |
| | *p*-value | 0.007 | 0.051 | 0.004 | 0.074 | <0.001 | <0.001 | | | | | |
| Clean reads | 0 | 59974 B | 70842 | 43763 B | 48132 B | 62964 AB | 40468 B | 6401 | 0.040 | <0.001 | <0.001 | <0.001 |
| | 5 | 67089 ABab | 71377 ab | 63326 Ab | 75744 Aa | 69921 Aab | 75001 Aa | 1968 | 0.009 | | | |
| | 14 | 67377 AB | 69169 | 69527 A | 75848 A | 68202 A | 72667 A | 2016 | 0.146 | | | |
| | 45 | 73966 Aa | 68423 a | 60153 Ab | 73618 Aa | 53947 ABbc | 48486 Bc | 2315 | <0.001 | | | |
| | 90 | 71015 Aa | 62192 abc | 57832 Aabc | 66824 Aab | 50882 Bbc | 47072 Bc | 3807 | 0.006 | | | |
| | SEM | 2024 | 4567 | 4166 | 4066 | 3753 | 2236 | | | | | |
| | *p*-value | 0.006 | 0.641 | 0.015 | 0.011 | 0.017 | <0.001 | | | | | |
| Observed species | 0 | 632 Aa | 236 b | 121 Bb | 102 Cb | 135 Cb | 232 b | 40.4 | <0.001 | <0.001 | <0.001 | 0.003 |
| | 5 | 307 C | 184 | 143 B | 89.3 C | 138 C | 255 | 36.8 | 0.054 | | | |
| | 14 | 300 C | 253 | 176 B | 189 B | 212 BC | 257 | 22.1 | 0.077 | | | |
| | 45 | 326 C | 321 | 244 A | 315 A | 252 AB | 310 | 25.8 | 0.151 | | | |
| | 90 | 411 B | 292 | 291 A | 251 AB | 309 A | 340 | 34.8 | 0.244 | | | |
| | SEM | 22.2 | 40.8 | 1544 | 23.3 | 21.2 | 28.5 | | | | | |
| | *p*-value | <0.0001 | 0.2324 | <0.0001 | <0.0001 | 0.0007 | 0.1154 | | | | | |
| Shannon | 0 | 6.99 Aa | 2.72 b | 3.27 Bb | 2.62 Bb | 3.07 Bb | 4.46 ABb | 0.767 | 0.017 | <0.001 | <0.001 | 0.839 |
| | 5 | 4.74 Ba | 3.19 ab | 3.60 ABab | 1.36 Cb | 2.85 Bab | 3.56 Bab | 0.521 | 0.096 | | | |
| | 14 | 4.97 B | 4.31 | 3.70 AB | 2.68 B | 3.83 AB | 3.81 B | 0.334 | 0.057 | | | |
| | 45 | 4.90 B | 4.78 | 4.35 A | 3.76 A | 3.90 AB | 5.16 AB | 0.352 | 0.197 | | | |
| | 90 | 5.39 B | 4.52 | 4.52 A | 3.56 A | 4.82 A | 5.69 A | 0.384 | 0.076 | | | |
| | SEM | 0.199 | 0.724 | 0.217 | 0.271 | 0.268 | 0.405 | | | | | |
| | *p*-value | <0.001 | 0.263 | 0.011 | <0.001 | 0.003 | 0.019 | | | | | |

**Table 3.** *Cont.*

| Items | Day | H_D | H_L | I_H | I_T | S_T | S_S | SEM | *p*-Value | *p*-Value | | |
|---|---|---|---|---|---|---|---|---|---|---|---|---|
| | | | | | | | | | | A | T | A × T |
| Simpson | 0 | 0.976 A | 0.483 | 0.718 | 0.655 A | 0.750 BC | 0.871 AB | 0.116 | 0.140 | 0.009 | 0.005 | 0.956 |
| | 5 | 0.897 Ba | 0.665 ab | 0.841 a | 0.407 Bb | 0.691 Cab | 0.748 Ba | 0.075 | 0.051 | | | |
| | 14 | 0.920 Ba | 0.829 a | 0.842 a | 0.630 Ab | 0.814 ABa | 0.805 ABa | 0.037 | 0.013 | | | |
| | 45 | 0.893 B | 0.881 | 0.864 | 0.715 A | 0.835 AB | 0.916 A | 0.042 | 0.268 | | | |
| | 90 | 0.920 B | 0.856 | 0.860 | 0.725 A | 0.901 A | 0.933 A | 36.9 | 0.228 | | | |
| | SEM | 0.012 | 0.110 | 0.045 | 0.055 | 0.030 | 0.031 | | | | | |
| | *p*-value | 0.005 | 0.125 | 0.205 | 0.005 | 0.005 | 0.011 | | | | | |
| Chao1 | 0 | 648 Aa | 250 b | 124 Bb | 107 Cb | 139 Cb | 234 b | 40.6 | <0.001 | <0.001 | <0.001 | 0.002 |
| | 5 | 319 C | 193 | 149 B | 98.0 C | 145 C | 271 | 38.1 | 0.050 | | | |
| | 14 | 313 C | 262 | 186 B | 208 B | 219 BC | 271 | 23.6 | 0.117 | | | |
| | 45 | 340 C | 337 | 250 A | 336 A | 261 AB | 317 | 25.6 | 0.086 | | | |
| | 90 | 433 B | 301 | 297 A | 260 B | 316 A | 349 | | 0.221 | | | |
| | SEM | 23.4 | 40.6 | 17.1 | 26.2 | 22.0 | 29.5 | | | | | |
| | *p*-value | <0.001 | 0.205 | <0.001 | <0.001 | <0.001 | 0.128 | | | | | |

Whole-plant corn silages were processed at 6 locations in 3 areas of North China: Heilongjiang province (Daqing city (H_D; 124°43′42.074″ E, 46°18′32.083″ N) and Longjiang county (H_L; 123°7′32.120″ E, 47°21′23.396″ N), cold and wet agricultural area), Inner Mongolia Autonomous Region (Helin county (I_H; 111°36′21.625″ E, 40°28′47.672″ N) and Tumet Left Banner (I_T; 111°9′31.399″ E, 40°41′29.832″ N), temperate and dry pastoral area) and Shanxi province (Taigu (S_T; 112°37′53.792″ E, 37°25′57.230″ N) and Shanyin (S_S; 112°52′06.676″ E, 39°32′54.503″ N) counties, temperate and dry agricultural area). Sampling times were at 0, 5, 14, 45 and 90 days after ensiling. Note: SEM, standard error of the mean; A, sampling area; T, sampling time. Values with different uppercase [A–C] and lowercase [a–c] letters indicate significant differences among 6 sampling locations at the same time.

### 3.3. Correlation between Fermentation Quality and Bacterial Genera

pH had a negative correlation with *Lactobacillus* in all silages ($p < 0.05$). It correlated positively with *Weissella* in silages from Heilongjiang ($p < 0.05$), and with *Lactococcus*, *Leuconostoc* and *Weissella* in silages from Inner Mongolia and Shanxi ($p < 0.05$). *Lactobacillus* had a positive correlation with LA and AA contents in silages from Heilongjiang and Inner Mongolia ($p < 0.05$). LA concentration was negatively correlated with *Weissella* in silages from Heilongjiang ($p < 0.05$), with *Lactococcus* and *Leuconostoc* in silages from Inner Mongolia ($p < 0.05$), and with *Lactococcus*, *Leuconostoc* and *Weissella* in silages from Shanxi ($p < 0.05$). The AA content had a negative correlation with *Lactococcus* and *Leuconostoc* in silages from Inner Mongolia ($p < 0.05$), and with *Lactococcus* and *Weissella* in silages from Shanxi ($p < 0.05$) (Figure 5).

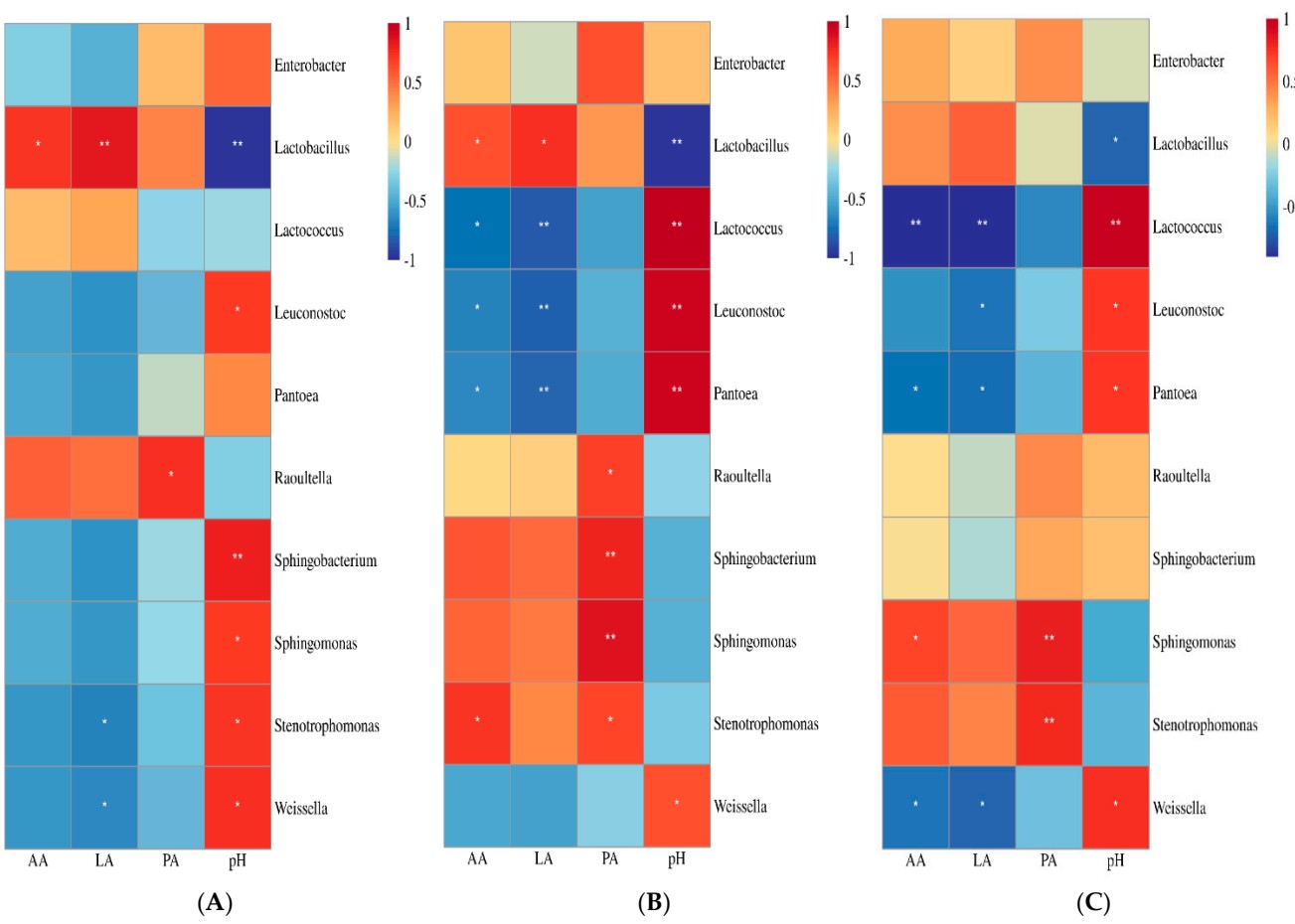

**Figure 5.** Correlation between the main bacterial genera (top 10) and fermentation quality (pH, lactic acid (LA), acetic acid (AA) and propionic acid (PA)) in silages from Heilongjiang (**A**; cold and wet agricultural area), Inner Mongolia (**B**; temperate and dry pastoral area), and Shanxi (**C**; temperate and dry agricultural area) during fermentation ($n = 30$).

## 4. Discussion

In the present study, whole-plant corn from Heilongjiang contained lower DM content (254 and 245 g/kg; Table 1), which might have been due to earlier harvesting (1/3 milk-line stage), and the cooler and wetter environment. This is similar to previous studies [2,12] that reported that the growing environment and harvesting stage influence the DM content of whole-plant corn. Earlier-harvesting corn contained lower DM content and greater amounts of water-soluble sugars, and its final silage had lower pH and greater lactic acid [23]. However, in the present study, silages from Inner Mongolia, with 285 and 288 g/kg of DM content, had a lower pH than that of other silages at 14 and 90 days, and less AN than that of silages from Heilongjiang at 45 and 90 days. In addition, the sampling

area did not affect LA concentration in the final silages (Table 1). This indicated that silages from Inner Mongolia had better satisfactory fermentation quality than that of other silages. Moreover, silages from Heilongjiang had higher AN than that of other silages (except for I_T) at 45 and 90 days, and contained the lowest DM content (Table 1). This indicated that the high moisture content in whole-plant corn before ensiling was not conducive to reducing AN content during fermentation. Whole-plant corn silage after 5 days of ensiling is in a stable fermentation phase according to Sun et al. [9]. The pH was less than 4.0 at 5 days and reached the lowest point at 60 days in all the silages except for H_D, which agreed with a previous study [9]; moreover, lactic acid reached a high level (>76 g/kg DM) at 5 days and increased to its peak point at 14 or 45 days; AN content increased to its highest point at 45 days (Table 1). Ferrero et al. [24] also reported reduced pH and increased contents of lactic acid and AN in whole-plant corn silage after 15 days of ensiling. Those mentioned-above indicated that, during the stable fermentation phase, fermentation did not stop, and fermentation parameters changed to some extent in whole-plant corn silage.

The main bacterial genera were *Leuconostoc*, *Weissella*, *Sphingobacterium* and *Stenotrophomonas* in H0, and *Leuconostoc*, *Lactococcus*, *Weissella* and *Pantoea* in I0 and S0 (Figure 3 and Supplementary Figure S1). H0 contained less *Lactococcus* and more *Sphingobacterium*, *Stenotrophomonas* and *Sphingomonas* than I0 and S0 (Supplementary Figure S2). Additionally, the bacterial community in I0 and S0 flocked together (except for S0_S_1) and separated from H0_D, H0_L_2 and H0_L_3 according to PCA (Supplementary Figure S3). This suggested that whole-plant corn from Inner Mongolia and Shanxi had a similar epiphytic bacterial community and differed from those from Heilongjiang. The different epiphytic bacterial community in raw materials might have resulted from the different geographical location [2,12,25]. Sampling locations in Inner Mongolia and Shanxi belong to a temperate continental climate and are close to each other; sampling locations in Heilongjiang, on the other hand, locate in temperate monsoon climates and are far from the other sampling locations (Figure 1). Guan et al. [2] and Gharechahi et al. [12] also reported the unique bacterial community in whole-plant corn from different sampling sites. The abundance of *Lactobacillus* was 1.02%, 1.08% and 1.51% in H0, I0 and S0, respectively (Figure 3 and Supplementary Figure S1); the same results were reported by previous studies [2,11–15,24,26]. This generally indicated that *Lactobacillus* is a minor taxon in whole-plant corn before ensiling. Additionally, the main LAB genera in fresh forages were *Leuconostoc*, *Weissella* and/or *Lactococcus* in the present study (Figure 3 and Supplementary Figure S1). This indicated that *Leuconostoc*, *Weissella* and *Lactococcus* might play a major role during the early stage (the first 24 h) of fermentation [9]. LAB counts in the raw materials were more than $10^6$ colony-forming units/g fresh weight in the present study, and similar results were reported by a previous study [9,24]. This suggests that LAB count in whole-plant corn before ensiling is sufficient for satisfactory fermentation quality in the final silage.

According to Sun et al. [9], after 3 days of ensiling, whole-plant corn silage is in a stable fermentation phase, during which *Lactobacillus* plays a key role among bacterial genera in silage. In the present study, *Lactobacillus* had greater abundance than that of other bacterial genera in silages with low pH (<4.0) from 5 to 90 days (Figure 3 and Supplementary Figure S1). Results were consistent with those of previous studies [9,11,12,14,27]. Moreover, Xu et al. [26], Drouin et al. [28] and Wang et al. [29] revealed that *Lactobacillus* was the dominant bacterial genus in the final whole-plant corn silage. This showed that *Lactobacillus* usually dominates bacterial community succession in whole-plant corn silage during fermentation. However, after 5 days of ensiling, the abundance of *Lactobacillus* was reduced (Figure 3 and Supplementary Figure S1), and LAB count began to decrease in all silages (Table 2). Sun et al. [9] also observed the reduced abundance of *Lactobacillus* and decreased LAB count during the stable phase in whole-plant corn silage without any inoculant. This suggested that the epiphytic *Lactobacillus* on whole-plant corn might have weak acid resistance. *Weissella* is the most important LAB genus in whole-plant corn silage from 5 to 24 h after ensiling [9,14]. It is an obligatory heterofermentative LAB species that converts water-soluble sugars into LA and AA during the early stage of fermentation

in silage [30]. In the present study, no *Weissella* was detected in I_T during the ensiling process (Figure 3), which might have resulted in the lower AA concentration in I_T than that in other silages at 5, 45 and 90 days (Table 1). *Sphingobacterium, Stenotrophomonas, Sphingomonas, Enterobacter, Pantoea, Raoultella, Klebsiella, Serratia* and *Rahnella* are enterobacteria as facultatively anaerobic Gram-negative bacteria [31], some of which are undesirable in silage [23]. In the present study, the total abundances of those genera rapidly decreased in the first 5 days, and then increased to 12.9%, 9.33% and 24.9% in H90, I90 and S90, respectively (Supplementary Figure S1). Those above-mentioned suggested that it is necessary to add inoculants with greater capacity for acid production and resistance during the ensiling of whole-plant corn, especially *Lactobacillus* inhibiting enterobacteria.

In the present study, *Pediococcus* had considerable abundance in I5_H (12.5%), while it was a minor taxon in other silages (<0.5%), which might have been the cause of the bacterial community in I5_H being separated from other silages at 5 days (Figure 2A). According to PCA, the bacterial community in silages from Shanxi began to separate at 14 days, and separated clearly from other silages at 45 and 90 days (Figure 2). Moreover, the bacterial community in silages from Heilongjiang and Inner Mongolia clustered together as the ensiling process (Figure 2). The above-mentioned results indicated that silages from Heilongjiang and Inner Mongolia had a similar bacterial succession pattern during the fermentation process and differed from the silages from Shanxi. Additionally, compared with silages from Heilongjiang and Inner Mongolia, silages from Shanxi had a higher reducing rate of *Lactobacillus* from 14 to 90 days (Figure 3 and Supplementary Figure S1), and contained less *Lactobacillus* as the dominant bacterial genus (Figure 4). This suggested that *Lactobacillus* in silages from Shanxi had weaker acid resistance than that in other silages, which might have contributed to the different bacterial succession pattern in whole-plant corn silages among sampling areas. Silva et al. [32] also reported a tendency of bacterial communities to cluster together in whole-plant corn silage during the ensiling process. However, Xu et al. [11] and Keshri et al. [14] reported that the bacterial community in final silages clearly separated from silages at other sampling times.

Sun et al. [9] reported that *Lactobacillus* in whole-plant corn silage dominates the bacterial community during the stable phase and correlates negatively with pH from 1 to 60 days after ensiling. In the present study, *Lactobacillus* dominated the bacterial community during fermentation and had a negative correlation with pH in all silages and a positive correlation with LA and AA concentrations in the silages from Heilongjiang and Inner Mongolia (Figure 5). In addition, *Lactobacillus* also correlated positively with LA and AA without reaching the significance level in the silages from Shanxi (Figure 5C). The final silages had lower pH (from 3.52 to 3.80) and AN content (from 36.6 to 91.4 g/kg TN), and higher LA content (from 73.0 to 139 g/kg DM) (Table 1). Guan et al. [2] also found a positive correlation between LA content and *Lactobacillus* in whole-plant corn silages collected from five sampling sites in Southwest China. Previous studies reported that *Lactobacillus* dominated the bacterial community during fermentation in silages with good fermentation quality [11–14,17,33]. This indicated that the activity of *Lactobacillus* in silage mainly contributes to forming and maintaining satisfactory fermentation quality in silage during the ensilage process. In the present study, *Leuconostoc, Lactococcus* and *Weissella* had a positive correlation with pH, and were negatively correlated with LA and AA in silages (except for *Lactococcus* as a minor taxon in silages from Heilongjiang). Sun et al. [9] reported that *Leuconostoc, Lactococcus* and *Weissella* in whole-plant corn silage had an important effect on bacterial succession and the reduced pH in the first 24 h of ensiling, but positively correlated with pH from 1 to 60 days as minor taxa. This suggested that *Leuconostoc, Lactococcus* and *Weissella* might be inhibited under anaerobic and acidic environments, and played a limited role in whole-plant corn silage after 5 days of fermentation. Correlations of *Stenotrophomonas, Sphingomonas, Enterobacter* and *Pantoea* with fermentation quality were similar in silages from Inner Mongolia and Shanxi, which differed from the silages from Heilongjiang (Figure 5). The reason for the different correlations might be that the final silages from Inner Mongolia and Shanxi contained a lower abundance of LAB genera

than that of their materials and had similar changes in abundance of those genera during fermentation (Supplementary Figure S1). Ogunade et al. [34] found a negative correlation of *Pantoea*, *Pseudomonas*, *Sphingomonas* and *Stenotrophomonas* with pH in alfalfa silage. The effect of those bacterial genera on fermentation quality of silage needs to be further studied.

## 5. Conclusions

The geographic growth location mainly impacted the epiphytic bacterial community on whole-plant corn. Whole-plant corn silages had satisfactory fermentation quality. *Lactobacillus* was a minor taxon in fresh forage and dominated the bacterial community in whole-plant corn silages during the fermentation process. The acid resistance of *Lactobacillus* in whole-plant corn silage determined the bacterial succession pattern during fermentation, and silages from Heilongjiang and Inner Mongolia had similar bacterial succession pattern. The activity of *Lactobacillus* during the ensilage process contributed to forming and maintaining a satisfactory fermentation quality in whole-plant corn silage.

**Supplementary Materials:** The following are available online at https://www.mdpi.com/article/10.3390/pr9050900/s1, Figure S1: Relative abundance of bacterial communities (genus level) in whole-plant corn silage from Heilongjiang (H; cold and wet agricultural area), Inner Mongolia (I; temperate and dry pastoral area) and Shanxi (S; temperate and dry agricultural area) at 0, 5, 14, 45 and 90 days after ensiling, Figure S2: Difference in bacterial communities (genus level) among whole-plant corn from Heilongjiang (H; cold and wet agricultural area), Inner Mongolia (I; temperate and dry pastoral area) and Shanxi (S; temperate and dry agricultural area), Figure S3: Principal component analysis of a bacterial community in whole-plant corn. Corn grown at six locations in three areas of Northern China: Heilongjiang province (Daqing city (H_D; 124°43′42.074″ E, 46°18′32.083″ N) and Longjiang county (H_L; 123°7′32.120″ E, 47°21′23.396″ N), cold and wet agricultural area), Inner Mongolia Autonomous Region (Helin county (I_H; 111°36′21.625″ E, 40°28′47.672″ N) and Tumet Left Banner (I_T; 111°9′31.399″ E, 40°41′29.832″ N), temperate and dry pastoral area) and Shanxi province (Taigu (S_T; 112°37′53.792″ E, 37°25′57.230″ N) and Shanyin (S_S; 112°52′06.676″ E, 39°32′54.503″ N) counties, temperate and dry agricultural area), Figure S4: Principal component analysis of a bacterial community in whole-plant corn silages from Heilongjiang province (Daqing city (H_D; 124°43′42.074″ E, 46°18′32.083″ N) and Longjiang county (H_L; 123°7′32.120″ E, 47°21′23.396″ N), cold and wet agricultural area), Inner Mongolia Autonomous Region (Helin county (I_H; 111°36′21.625″ E, 40°28′47.672″ N) and Tumet Left Banner (I_T; 111°9′31.399″ E, 40°41′29.832″ N), temperate and dry pastoral area) and Shanxi province (Taigu (S_T; 112°37′53.792″ E, 37°25′57.230″ N) and Shanyin (S_S; 112°52′06.676″ E, 39°32′54.503″ N) counties, temperate and dry agricultural area) at 0, 5, 14, 45 and 90 days after ensiling.

**Author Contributions:** Conceptualisation, C.W., H.H. and Y.X.; methodology, C.W., H.H. and L.S.; software, L.S., N.N. and S.C.; validation, C.W., H.H. and Y.X.; visualisation, S.C.; formal analysis, L.S. and N.N.; investigation, C.W., H.H., L.S. and N.N.; resources, C.W. and H.H.; data curation, C.W., H.H. and H.X.; writing—original draft, C.W. and H.H.; writing—review and editing, C.W., H.H., H.X., Y.J. and Y.X.; supervision, Y.X.; project administration, Y.X.; funding acquisition, Y.X. All authors have read and agreed to the published version of the manuscript.

**Funding:** This study was funded by the National Key R&D Program of China (grant number 2017YFE0104300) and the Science and Technology Project of Inner Mongolia (grant number 2020GG0049).

**Institutional Review Board Statement:** Not applicable.

**Informed Consent Statement:** Informed consent was obtained from all subjects involved in the study.

**Data Availability Statement:** The data presented in this study are available on request from the corresponding author.

**Conflicts of Interest:** The authors declare no conflict of interest.

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
