# Peer review of "Bacterial Succession Pattern during the Fermentation Process in Whole-Plant Corn Silage Processed in Different Geographical Areas of Northern China"

_processes, doi:10.3390/pr9050900_

Round 1

Reviewer 1 Report

  1. Line 28 – Change shanxi to Shanxi
  2. Line 28 – the use of lower and greater seems to me less correct than the use of less or more instead of lower and greater, respectively: … contained lower Lactobacillus and greater Leuconostoc … contained less Lactobacillus and more Leuconostoc
  3. Lines 28-29 – I cannot understand the sentence … and had separated bacterial community from 14 to 90 d. Separated?
  4. Line 36 – “during fermentation process” as keyword it is not important considering the ones firstly pointed.
  5. Lines 45-46 – consider to remove “contribute to use it as important and popular feedstuff for dairy production, which” from the sentence that starts as “The characteristics of whole-plant corn silage …”. The idea is presented in the beginning and in the end of the paragraph and in this sentence it is only a repeated idea and cuts off the description of the characteristics of whole-plant corn silage.
  6. Line 60 – Change LAB to lactic acid bacteria (LAB); the first time mentioned in text.
  7. Line 64 – Change lactic acid bacteria (LAB) to LAB; the second time mentioned in text.
  8. Line 65 – fermentation relay existing??
  9. Lines 82-86 – Consider the use of [( )] to easier the reading of all the information about locations.
  10. “… Heilongjiang Province [Daqing City (H_D) and Longjiang County (H_L), cold and wet, agriculture area], Inner Mongolia Autonomous Region [Helin County (I_H) and Tumet Left Banner (I_T), temperate and dry, pasture area], and Shanxi Province [Taigu County (S_T) and Shanyin County (S_S), temperate and dry, agriculture area] …” Consider it here and all times mentioned in the text, Tables and Figures legends.
  11. Line 102 – Change the sentence to: The pH of silage was detected using a pH meter (PB-10, Sartorius, Gottingen, Germany) to measure the silage extract.
  12. Line 103 – Here and all times you mention that bacteria are counted it is better to say “other bacteria” instead of “bacteria” since LAB and coliforms are also bacteria. In fact in nutrient agar it is expected that coliforms will also grow.
  13. Lines 109-111 – Use [( )]: “… [detector, SPD-20A diode array detector (210 nm); column, Shodex RS Pak 109 KC-811 (50 °C, Showa Denko K.K., Kawasaki, Japan); mobile phase, 3 mM HClO4 (1.0 110 ml/min)] …”
  14. Line 156 – Change 5669279 to 5,669,279
  15. Line 158 – Change “affected observed species, shannon, simpson, and chao1” to … affected the observed species diversity, shannon, simpson, and chao1 indexes …
  16. Tables 1,2,3 description – In description: SERM, standard error of the mean BUT in table SEM.
  17. Tables 1,2,3 description – sampling times of each lactation?
  18. Figure 3 - The italic format is missing in the genera names.
  19. Line 227 – positively with Lactococcus … according to the Figure it is NEGATIVELY
  20. Figure 4 – These graphs are not readable. They represent huge information but since are impossible to read must be improved.
  21. Line 267 – Change maim to main
  22. °C – Review the use of °C; it is presented with and without a space (e.g. 65 °C, 105°C) and it should be presented according journal specifications.
  23. Line 418 – Reference 20 – Apply the italic format to Enterococcus
  24. Line 436 – Reference 27 – Change micro-organisms to microorganisms
  25. Line 446 – Reference 30 – Apply the italic format to Escherichia coli

Author Response

Dear Reviewer,

Thanks for your comments about the manuscript.

The attachment is the responses, please check it.

The manuscript has been edited by English speaker.

All the best,

Yanlin Xue

Reviewer 2 Report

In abstract 

On basis of which criterion you said that the fermentation quality is good?

Do you do the sensory analysis 

2. Materials 

Please give the latitude and longitude for each region where the samples have been collected 

2 Analysis 

Please separated

2.1 Physicochemical analysis

2.2 Microbial analysis

2.2.1 Microbial load 

2.2.2 Microorganisms molecular identification 

Results

Microorganisms identification: please give the name species but not the genera.

The paper need the major revision

Author Response

Dear Reviewer,

Thanks for your comments about the manscript.

The point-by-point response toyour comments is attached.

All the best,

Yanlin Xue

Reviewer 3 Report

The manuscript aimed to evaluate the changes in the bacterial community during fermentation process in whole-plant corn silages processed in three area of China. The authors performed a good work and the manuscript is well organized.

However, I would suggest some clarification/improvement.

L39-43: I agree with you, however there is need to specify that the sentence is true when talking about well fermented silages

L54: ensiling process

INTRODUCTION: there are recent paper about temperature, ensiling duration and microbial succession. I suggest to consider here or in the discussion section: doi:10.3390/microorganisms7120595, https://doi.org/10.1002/mbo3.1153, doi:10.1111/1751-7915.13623, https://doi.org/10.3168/jds.2020-18733

L90: how it was chopped?

L91: what is the weight of each plastic bag?

L103: The counts of LAB, ENTEROBACTERIA, TOTAL AEROBIC BCTERIA, and yeast… change this throughout the manuscript

L113: add reference

L117-118: according to manufacturing instruction

DISCUSSION: I suggest to improve the discussion about the effect of DM on silage fermentation and the effect of time of conservation on microbial community (already discussed) and fermentation.

I suggest to consider doi:10.3390/microorganisms7120595, https://doi.org/10.1002/mbo3.1153, doi:10.1111/1751-7915.13623, https://doi.org/10.3168/jds.2020-18733

Author Response

Dear Reviewer,

Thanks for your comments about the manuscript.

The attachment is the response to your comments.

Please check it.

All the best,

Yanlin Xue

Round 2

Reviewer 2 Report

For microbiology study, particularly on microbial identification it is important and interesting to give the species  name but not the genera.  Indeed, some species can to be used as starters for example. For your next studies, you can do the PCR-RLFP and sequencing to have the name of species for example. Also, for the next time, use the simple figure to present your results.

Conclusion

I accept that this paper for publication

Best regards

Reviewer 3 Report

The authors improved the quality of manuscript